# New Substituted Benzoylthiourea Derivatives: From Design to Antimicrobial Applications

**DOI:** 10.3390/molecules25071478

**Published:** 2020-03-25

**Authors:** Carmen Limban, Mariana Carmen Chifiriuc, Miron Teodor Caproiu, Florea Dumitrascu, Marilena Ferbinteanu, Lucia Pintilie, Amalia Stefaniu, Ilinca Margareta Vlad, Coralia Bleotu, Luminita Gabriela Marutescu, Diana Camelia Nuta

**Affiliations:** 1Department of Pharmaceutical Chemistry, “Carol Davila” University of Medicine and Pharmacy, 020956 Bucharest, Romania; carmen.limban@umfcd.ro (C.L.); ilinca.vlad@drd.umfcd.ro (I.M.V.); diana.nuta@umfcd.ro (D.C.N.); 2Department of Microbiology, Faculty of Biology & Research Institute of the University of Bucharest (ICUB), University of Bucharest, 060101 Bucharest, Romania; lumi.marutescu@gmail.com; 3The Organic Chemistry Center of Romanian Academy “C. D. Neniţescu”, 060023 Bucharest, Romania; dorucaproiu@gmail.com (M.T.C.); fdumitra@yahoo.com (F.D.); 4Inorganic Chemistry Department, Faculty of Chemistry, University of Bucharest, 020462 Bucharest, Romania; marilena.cimpoesu@g.unibuc.ro; 5National Institute of Chemical-Pharmaceutical Research & Development, 031299 Bucharest, Romania; lucia.pintilie@gmail.com (L.P.); astefaniu@gmail.com (A.S.); 6Stefan S. Nicolau Institute of Virology, Mihai Bravu 285, Bucharest, 030304, Romania; cbleotu@yahoo.com

**Keywords:** benzoylthiourea derivatives, antimicrobial, antibiofilm, X-ray, in silico

## Abstract

The increasing threat of antimicrobial resistance to all currently available therapeutic agents has urged the development of novel antimicrobials. In this context, a series of new benzoylthiourea derivatives substituted with one or more fluorine atoms and with the trifluoromethyl group have been tested, synthesized, and characterized by IR, NMR, CHNS and crystal X-ray diffraction. The molecular docking has provided information regarding the binding affinity and the orientation of the new compounds to *Escherichia coli* DNA gyrase B. The docking score predicted the antimicrobial activity of the studied compounds, especially against *E. coli*, which was further demonstrated experimentally against planktonic and biofilm embedded bacterial and fungal cells. The compounds bearing one fluorine atom on the phenyl ring have shown the best antibacterial effect, while those with three fluorine atoms exhibited the most intensive antifungal activity. All tested compounds exhibited antibiofilm activity, correlated with the trifluoromethyl substituent, most favorable in para position.

## 1. Introduction

In the last decades, thiourea derivatives have received considerable attention owing to their effective biological activities, including the antibacterial [1,2,3,4,5,6], antitubercular [7,8], antimalarial [9,10,11], antileishmanial [12], antifungal [13,14,15,16], or antiviral effects [17,18,19,20,21,22,23].

Thiourea substitution in anacardic acids’ C-8 alkyl chain provided anti-*Escherichia coli*, *Pseudomonas aeruginosa*, *Staphylococcus aureus*, and *Streptococcus pyogenes* activity. Methyl 2-methoxy-6-{8-{3-[3-(trifluoromethyl)phenyl]thioureido}octyl}benzoate exhibited a biological activity similar to the standard antibiotic ampicillin [24]. On the contrary, *p*-(3-trifluoromethyl-5-substituted-pyrazol-1-yl)benzenesulfonylthiourea derivatives have been less effective than ampicillin and griseofulvin against *S. aureus*, *E. coli*, *Aspergillus niger* and *Candida albicans* [25]. Thereby, the (*E*)-*N*-[4-(benzamidomethylenamino)phenylcarbamothioyl]benzamide exhibited a good antibacterial activity [4].

The *N*-[(arylamino)thioxomethyl]-3,5-dichlorobenzo[*b*]thiophene-2-carboxamide displayed in vitro both antibacterial (i.e., against *E. coli*, *Proteus vulgaris*, *Bacillus megaterium*, *S. aureus*) and antifungal (i.e., against *A. niger*) activities [5]. Also, a series of diversely substituted *N*-*p*-methylbenzoyl-*N′*thiourea derivatives, such as *N*-(4-methylbenzoyl)-*N′*-(4-chloro-2-nitrophenyl)thiourea and *N*-(4-methylbenzoyl)-*N*′-(4-methylphenyl)thiourea have been proven to exhibit antibacterial and antifungal activities [6] (Figure 1).

Thiophanate and thiophanate methyl are thiourea-based fungicides largely used on different fruits and vegetable crops. The 1,3-disubstituted aliphatic and aromatic symmetrical and unsymmetrical thioureas proved to be effective against *Pyricularia oryzae* and *Drechslera oryzae* phytopathogens [26], while *N*-(*o*-fluorophenoxyacetyl)thiourea derivatives exhibited herbicidal activity against *Amaranthus retroflexus* L. [27].

In the last 15 years the fluoro-derivatives class has flourished. In 2018 FDA approved 17 new molecules, among which fostamatinib, baloxavir, marboxil, doravirine, ivosidenib, and apalutamide [28]. The isosteric modification, consisting of replacing hydrogen with fluorine, modifies the steric and electronic properties of the resulting molecules, affecting their metabolic transformations because of the relative stability given by the C-F energy bond (485.7 kJ/mol) [29]. The presence of the fluorine atom modulates the pharmacodynamic and pharmacokinetic properties [25]. Drug fluorination is commonly used to modulate pKa values, to enhance membrane permeation, and to increase liposolubility and, thus, their in vivo absorption and transport [30,31]. The thiourea skeleton favors other chemical modulations (e.g., incorporation of fluorine), these derivatives being thus promising candidates, both as active molecules and ligands. For instance, the fluoroquinolone derivatives having thioureido moiety and piperazin-1-yl groups have shown a significant improvement of the activity against Gram-positive (*S. aureus*, *B. subtilis*) and Gram-negative (*E. coli*, *P. aeruginosa*) bacterial strains [32]. The 2-(6’-fluorobenzothiazol-2’-ylamino)-4,6-(disubstituted thioureido)-1,3-pyrimidine derivatives and the 1-(isomeric fluorobenzoyl)-3-(isomeric fluorophenyl)thiourea compounds have shown inhibitory activity against the above-mentioned four species, correlated with the presence of nitro groups and with the presence and position of fluorine substituent on aroyl and aryl rings [33,34]. The last ones exhibited a better antifungal activity against *Rhizopus oryzae*, *Aspergillus terreus*, *Fusarium oxysporum*, *A. niger*, and *A. fumigatus*, probably due to the presence of fluorine, which increased their lipophilicity, penetration rate, and intracellular transport to the microbial target. An improved receptor binding could be achieved due to the higher polarizability assured by the C—F bond [34].

On the other hand, the trifluoromethyl group is an important substituent in many drugs or agrochemical compounds because it increases the lipophilicity and metabolic stability [29]. Therefore, a number of synthesis methods for such compounds have been developed [35] so that it can be expected that in the future a lot of new molecules with biological activity from the class of trifluoromethyl derivatives will be developed [36].

Furthermore, the synthesis methods of organofluorine chemicals are intensively studied [37] due to their various applications in medicine, chemistry, and agriculture. Organofluorine compounds are used for treating bacterial (e.g., fluoroquinolones), fungal (fluconazole, flucytosine), and viral (e.g., clevudine for hepatitis B, efavirenz for HIV infection) infections, malaria (mefloquine), cancer (e.g., 5-fluorouracil, a thymidylate synthase inhibitor, floxuridine, an antimetabolite in the treatment of colorectal cancer, bicalutamide, a nonsteroidal anti-androgen, flutamide with anti-androgen properties), diabetes (tolrestat, an aldose reductase inhibitor used in diabetic complications, sitagliptin, a dipeptidyl peptidase-4 inhibitor), superficial or systemic hypercholesterolemia (atorvastatin, an inhibitor of 3-hydroxy-3-methyl-glutaryl-CoA reductase, torcetrapib, a cholesterylester transfer protein inhibitor), asthma and allergic rhinitis (fluticasone propionate), glaucoma (travoprost, tafluprost), and depression (e.g., fluoxetine, a selective serotonin reuptake inhibitor), as well as for maintenance of general anesthesia (desflurane).

The common isotope of fluorine (^19^F) is the favorite heteroatom for incorporation into drugs [26], while the isotope (^18^F) is used for radiolabelling of some ligands used in HD PET imaging for Huntington disease diagnosis [38]. Recently, fluorodopa (FDOPATM), the fluorine-18 isotopologue of L-DOPA, was approved as a radiotracer in PET for the visualization of nerve cells in patients with symptoms of Parkinson disease [39].

Presently, the emergence of antimicrobial resistance in human, animal, and plant pathogens, correlated with the unavailability of newer drugs challenges the efficiency of current antimicrobial agents [40]. Therefore, comprehensive and multidisciplinary efforts across human medicine, veterinary, agricultural, and environmental sectors are needed to reduce this global threat by investigating the mechanisms of resistance, improving antimicrobial stewardship, and developing antimicrobial agents [41,42,43].

Our aim was to exploit the multiple advantages of 2-((4-ethylphenoxy)methyl)-*N*-(fluoro/trifluoro-substituted phenylcarbamothioyl)benzamides to obtain potent antibacterial and antifungal agents, potentially useful for both clinical and agricultural sectors.

## 2. Results

As a continuation of our previous research [44,45], we have synthesized new compounds bearing the fluorine atom/trifluoromethyl group and the thiourea moiety (i.e., 2-((4-ethylphenoxy)methyl)-*N*-(fluoro/trifluoromethyl-phenylcarbamothioyl)benzamides), and evaluated their antimicrobial activity.

### 2.1. Chemistry

The new compounds, **5a**–**g**, were obtained with good yields using the synthesis route presented in Scheme 1.

Phthalide was refluxed with potassium *p*-ethylphenoxide in xylene, resulting in the potassium salt of 2-(4-ethylphenoxymethyl)benzoic acid (**1**), which was separated from xylene due to its solubility in the 10% aqueous potassium hydroxide solution. The 2-(4-ethylphenoxymethyl)benzoic acid (**2**) was precipitated using 10% HCl solution. The 2-(4-ethylphenoxymethyl)benzoyl chloride (**3**) was prepared by refluxing acid **2** with thionyl chloride, in anhydrous 1,2-dichlorethane.

The reaction between compound **3** with ammonium thiocyanate in dry acetone resulted in in situ 2-(4-ethylphenoxymethyl)benzoyl isothiocyanate (**4**). The appropriate primary aromatic amines were directly added to the reaction mixture, in dry acetone, resulting in new derivatives **5a–g**.

The obtained compounds were solid, white, or light-yellow crystals, exhibiting solubility in chloroform and acetone at room temperature, soluble in lower alcohols, benzene, toluene, and xylene after heating and insoluble in water.

The melting point, elemental analysis, infrared (IR), and nuclear magnetic resonance (NMR) spectral studies were used for the characterization and structure confirmation of the new compounds.

In the IR spectra, the stretching bands due to νN-H of the amide group can be found to the highest values of the wave numbers. These were sharp peaks with a medium intensity occurring in the region 3401 cm^−1^. The thioamide group showed a less intense stretching band at 3155–3183 cm^−1^ and, with a high probability, the band located at 1374–1392 cm^−1^ can be attributed to the thioamide group. For the observed antisymmetric stretching vibrations, methyl and methylene groups gave saturated νC-H stretching bands at about 2959–2970 cm^−1^ and 2912–2933 cm^−1^, respectively; These bands are typical for aromatic compounds containing saturated carbons. A medium or intense sharp stretching band, shown in the IR spectrum of these compounds in the region 1676–1695 cm^−1^ is due to the νC=O vibrations. The νC=O band was lower than that observed with the ordinary carbonyl absorption (1730 cm^−1^). This low frequency value can be explained as a result of conjugated resonance with the phenyl ring and the formation of intramolecular hydrogen bonding with N-H. Near to this peak lies the intense band of δN-H, with a maximum at 1506–1514 cm^−1^, which overlaps the aromatic core vibrations. These compounds also show a typical alkyl-aryl ether at 1237–1256 cm^−1^ for the antisymmetric vibration, and 1013–1045 cm^−1^ for the symmetric one. The presence of fluorine in the new compounds is proven by the stretching bands situated at 1013–1079 cm^−1^.

The structure of the new compounds was also confirmed by the NMR spectra. The new compounds were dissolved in DMSO-d6 (hexadeuterio-dimethyl sulfoxide) and the chemical shifts’ values, expressed in parts per million (ppm), were referenced downfield to tetramethylsilane for ^1^H-NMR and ^13^C-NMR and upfield to trichloro-fluoro-methane for ^19^F-NMR and the constants (*J*) values in Hertz. The chemical shifts for hydrogen and carbon atoms were established also by GCOSY, GHMBC, GHSQC experiments. The ^1^H-NMR data were reported in the following order: Chemical shifts, multiplicity, coupling constants, number of protons, and signal/atom attribution. Spin multiplets were given as s (singlet), d (doublet), t (triplet), q (quartet), m (multiplet), dd (double doublet), dt (double triplet), dq (double quartet), ddd (doublet of a double doublet), td (triple doublet), tdd (triplet of a double doublet), and br (broad) signal. The ^13^C-NMR data were reported in the following order: chemical shifts, coupling constants in some cases, and signal/atom attribution ((Cq) quaternary carbon). The chemical shift of N-H protons was found in the range of δ 12.14–12.59 ppm for hydrogen bonds, and other protons of the N-H group were found in the range of δ 11.56–12.14 ppm. The hydrogen atom of the N-H group was observed downfield because of the presence of resonance and hydrogen bonds. The ^13^C-NMR signals of the carbonyl group, appearing at δ 169.75–170.61 ppm were due to the existence of the intramolecular hydrogen bond related to the carbonyl oxygen atom. The carbon atom of the thiocarbonyl group at δ 179.06–182.27 ppm showed the highest values.

The molecular structure of **5d** was analyzed by single crystal X-ray diffraction method, and crystallographic data are: C_23_H_19_F_3_N_2_O_2_S (Mr = 444.47 g mol^−1^), triclinic, *a* = 7.1369(2) Å, *b* = 11.8876(3) Å, *c* = 13.4640(9) Å, *α* = 102.869(7)°, *β* = 91.557(6)°, *γ* = 104.241(7)°, *V* = 1075.31(10) Å^3^, T = 293 K, space group P-1 (#2), Z = 2 (Appendix A). The molecular structure (Appendix A) shows that 2-(4-ethylphenoxymethyl)benzoyl fragment is bonded to the 2,4,6-trifluorophenyl thiourea moiety. The C=O and C=S bonds are oriented in a trans configuration in a plane stabilized by an intramolecular hydrogen bond N2-H···O2 (Figure 2). This plane forms a dihedral angle of 52.58° with the central phenyl ring. This characteristic appears in similar structures involving the benzoyl fragment connected to a thiourea fragment, as was seen by searching the Cambridge Crystallographic Data Centre (CCDC) (e.g., ethyl 4-[3-(2-methylbenzoyl)thioureido]benzoate (FUGZOX code in CCDC) [46], or N-(2-methylbenzoyl)-N′-(4-nitrophenyl)thiourea (GOJNAV code in CCDC) [47]. The 4-ethylphenoxymethyl fragment forms with central phenyl ring a 30.85° dihedral angle. The 2,4,6-trifluorophenyl ring forms with previous discussed plan another dihedral angle of 66.75°. Bond distances and angles are listed in the Appendix A. The unit cell contains two asymmetric units. The crystal packing for **5d** (see Appendix A) is based on hydrogen bonds’ supramolecular interactions, implying the 2,4,6-trifluorophenyl fragment belongs to neighboring molecules, being at the same time hydrogen acceptor and donor. The C-H···π supramolecular interactions involve the ethyl and phenyl ring from the 4-ethylphenoxymethyl fragments from different molecules (Figure 3).

### 2.2. Computational Studies

For the structure–activity relationship (SAR) studies, the following electronic properties were determined: Highest occupied molecular orbital (HOMO) and lowest unoccupied molecular orbital (LUMO) energy values, HOMO and LUMO orbital coefficients distribution, molecular dipole moment (for **5d** compound 5.15 debye), polar surface area (for **5d** compound PSA = 34.930 Å^2^) (a descriptor that has been shown to correlate well with passive molecular transport through membranes and, therefore, allows thar predictions for the membrane transport properties), the ovality (for **5d** compound 1.62), polarizability (for **5d** compound 74.42 10^−30^ m^3^), and the octanol water partition coefficient (for **5d** compound logP = 6.29) [48].

Properties relating to 3D structure of **5d** compound, bond lengths and angles, are presented in correlation with the experimental data (Appendix A). NMR spectra of **5d** compound have been calculated with Spartan 14 software. The experimental and calculated spectra proved to be correlated (Appendix A).

#### 2.2.1. Frontier Molecular Orbitals Analysis

Frontier molecular orbitals (FMOs) (the most important being HOMO and LUMO) analysis are predicting for the chemical stability and interactions between atoms and dictating the optical properties and biological activities of a certain molecule. HOMO indicates one molecule’s capacity to donate, while LUMO indicates one molecule’s capacity to accept an electron [49,50]. The HOMO and LUMO, calculated at the B3LYP/6-31G* level can be seen in Figure 4 for compound **5d**. The “blue and red” regions of the graphic correspond to the orbital’s positive and negative values. The frontier orbital gap (ΔE) characterizes the chemical reactivity. The high value of the HOMO-LUMO gap (ΔE) calculated for the **5d** compound refers to a chemically stable molecule [51].

#### 2.2.2. Molecular Electrostatic Potential (MEP)

MEP has been evaluated using the B3LYP method with the basic set 6-31G* to investigate the chemical reactivity of a molecule, indicating the reactive sites of nucleophilic or electrophilic attack in hydrogen-bonding interactions involved in the biological recognition process [52,53]. The putative electrostatic map of the compound **5d** shows hydrophilic regions in red (negative potential) and blue (positive potential) and hydrophobic regions in green (Figure 5a). The local ionization map is the result of the overlay of the energy of electron removal (ionization) on the electron density and can also indicate the electrophilic addition (Figure 5b). The LUMO map, representing a superposition of the absolute value of LUMO on the electron density indicates the nucleophilic addition (Figure 5c).

#### 2.2.3. Docking Studies

Molecular docking allowed us to establish an accurate prediction of the optimized conformation of new compounds (as ligands) and their target receptor protein in order to achieve a stable complex.

The score and hydrogen bonds formed with the amino acids from group interaction atoms are used to predict the binding modes, the binding affinities and the orientation of the docked substances (Figure 6a–i and Appendix A) in the active site of the protein receptor (Appendix A). The docking score used in the Drug Discovery Workbench is the PLANTS_PLP_ score [52].

The ligand orientation and position suggested by the molecular docking studies and the binding modes of ligands have been validated by redocking (Figure 6c,e,h, and Appendix A). According to the docking score evaluation, the best docking score was obtained for **5e**: −63.91 (RMSD 0.10). The compound **5e** shows three hydrogen bonds with ASN 45 (2.870 Å), ILE 78 (3.244 Å) and GLY 77 (2.924 Å) and its orientation is different from the co-crystallized one. The same orientation of the co-crystallized is shown by the the compounds **5b**, **5c**, **5d**, and **5f** (Figure 7a). The compounds **5a** and **5g** show also the same orientation as the **5e** compound (Figure 7b).

The results for **5d** compound revealed a docking score −56.80 (RMSD 0.20) and two hydrogen bonds with GLY 101 (3.144 Å and 3.381 Å) (Figure 6d).

The docking studies revealed that compounds **5e** (docking score −63.91, RMSD 0.22), **5a** (docking score −61.58, RMSD 0.05), **5g** (docking score −59.92, RMSD 0.24), **5b** (docking score-59.66, RMSD 0.46) and **5d** (docking score −56.81, RMSD 0.20 ) presented a good docking score (Appendix A). After correlating the in silico prediction with the experimental data, it was observed that all these four compounds displayed the same activity against *E. coli* (MIC: 128 µg/mL). Also, it has been observed that the compounds **5c** (docking score −57.73, RMSD 0.08) and **5f** (docking score −56.18, RMSD 0.22) displayed a lower activity against *E. coli* (MIC: > 256 µg/mL).

The number of hydrogen bond donors, the number of hydrogen bond acceptors, and log P were calculated and are presented in Table 1. The calculated parameters can predict if a molecule possesses properties that might turn it into an active drug, according to the Lipinski’s rule of five [53]: The number of hydrogen donors <5, the number of hydrogen acceptors <10, molecular weight <500 Da, and the octanol-water partition coefficient (log *P*) < 5. The log P calculation was performed using the XLOGP3-AA method [54].

### 2.3. Antimicrobial Activity Assay

The synthesized compounds were tested by a quantitative method, allowing us to establish the MIC (minimum inhibitory concentration) value (Table 2). The compounds **5c** and **5f** proved to have a low inhibitory effect against the tested strains. The compounds encoded **5a** and **5b** proved to be active against *E. coli* and *P. aeruginosa* strains, exhibiting the same MIC value. The compound **5d** exhibited both antibacterial and antifungal activity, being active on *E. coli* and *C. albicans* strains. The compound **5a** exhibited the largest spectrum of antibacterial activity, being active both against Gram-positive and −negative strains. Interestingly, the largest spectrum of activity of the compound **5a** was correlated with the presence of one fluorine atom as substituent on the phenyl ring.

Compared to the Gram-positive strains, the Gram-negative bacterial strains proved to be more susceptible to the tested compounds, with five of the seven tested compounds being active against the *E. coli* strain. *S. aureus* and *B. subtilis* proved to be resistant to all tested compounds, while *E. faecalis* and *C. albicans* were susceptible to only one compound, respectively, **5a** and **5d** (Table 2).

The presence of multiple fluorine atoms (e.g., three atoms in the compounds **5b**, **5c**, **5d**, and **5e**) located at different positions in the compounds did not correlate with the intensity of the antibacterial effect, with three of them exhibiting the same anti-*E. coli* activity similar to **5a**, with the other one displaying a lower activity against the same strain. All of them were inactive on *E. faecalis* and all but **5b** against *P. aeruginosa* when compared with **5a**. The compounds substituted with the trifluoromethyl group (**5e** and **5g**) were active on *E. coli*. Further, the presence of three fluorine atoms in the positions 2, 4, and 6 conferred antifungal properties for the compound **5d**. 

The antibiofilm activity spectrum was different from that obtained on planktonic strains, with all tested compounds being able to inhibit the formation of biofilms formed by at least two of the tested strains. The *E. coli*, *S. aureus*, and *B. subtilis* biofilms were the most recalcitrant to the tested compounds, with a maximum of three compounds exhibiting antibiofilm activity (Table 3). The *E. faecalis* and *C. albicans* biofilms were susceptible to the tested compounds, excepting **5c** and **5g**. The most active compounds on the development of microbial biofilms proved to be those substituted by the trifluoromethyl group, the most favorable substitution being in para position (Table 3).

## 3. Discussion

The emergence and spread of antimicrobial resistance have become some of the major worldwide health problems today, raising the acute need for developing novel antimicrobial agents. Considering the valuable pharmacological properties of thiourea derivatives, our aim was to design, synthesize and evaluate new compounds with fluorinated and trifluoromethylated aryl thiourea scaffolds. Fluorine substitution can improve the efficacy of drugs by influencing the absorption, tissue distribution, the rate and the route of biotransformation, the pharmacodynamic and the toxicological properties. Our interest was to optimize the substitution atoms or the functional groups present on the phenyl bound to nitrogen atom of thiourea in order to increase the antimicrobial activity of the obtained derivatives.

In this direction, we synthesized new 2-((4-ethylphenoxy)methyl)-*N*-(fluoro/trifluoromethyl-phenylcarbamothioyl)benzamides, and evaluated their antimicrobial activity.

The best antibacterial activity (against *E. coli*, *P. aeruginosa*, and *E. faecalis*) was recorded for the compounds bearing one fluorine atom, while the antifungal effect was favored by the isomeric substitution with three fluorine atoms. The **5d** exhibited the lowest recorded MIC value against the *C. albicans* fungal strain. This effect could be explained by the enhancement of lipid solubility and permeability of the fungal cell wall, with a particular structure owing to the eukaryotic nature of this microorganism, for the antimicrobial compound.

The spectrum of the antibiofilm activity was different from that recorded on planktonic cells, with all tested compounds inhibiting the ability of some microbial strains to form biofilms.The intensity of this activity was correlated with the presence of the trifluoromethyl substituent, most favorably in para position and the compound **5g** being the most active.

Despite the great susceptibility of planktonic *E. coli* cells to the majority of the tested compounds, the adherent *E. coli* cells were more resistant, remaining susceptible to only two compounds (**5e**, **5g**). This could account for the increased tolerance of microbial biofilms to the usual doses of antibiotics.

It is worth noting that *P. aeruginosa* biofilms were highly susceptible to the all tested compounds, except **5a**.

Some electronic and molecular properties were discussed in order to assess the flexibility and the binding ability of the studied conformer to bind to the receptor protein. The docking studies using *E. coli* DNA gyrase B revealed that all compounds showed a good docking score, indicating the inhibition of DNA replication as a potential mechanism of their antimicrobial activity that is worth to be studied further.

## 4. Materials and Methods

### 4.1. Chemistry

Highest quality phthalide, *p*-ethylphenol, ammonium thiocyanate, primary aromatic amines, and thionyl chloride used for the synthesis were purchased from Sigma-Aldrich Chemical Co. Analytical grade solvents (1,2-dichloroethane, xylene, isopropanol, acetone) supplied by Sigma- Aldrich were used and were dried when necessary. Potassium hydroxide and hydrochloric acid were obtained from Merck, and potassium carbonate from Fluka Chemical.

Acetone was dried over K_2_CO_3_ and then distilled. Ammonium thiocyanate was heated at 100 °C before use. The liquid amines were dried with potassium hydroxide and then distilled.

The reaction progress and the purity were investigated by thin layer chromatography on silica gel 60 F254 (0.5 mm thick) plates (Merck, Germany) and unidimensional migration using ethyl acetate/cyclohexane (4:6 *v*/*v*) as eluent with visualization by ultraviolet light (λ = 254 nm) and exposed to iodine vapors. 

Melting points were measured by heating a small amount of sample, by the capillary tube method, using an Electrothermal 9100 apparatus (Bibby Scientific Ltd., Stone, UK), without thermometer correction. The tube was observed with a magnifying glass, with a temperature range of 0.5–1.0 °C indicating a relatively high level of purity.

C, H, N, and S analyses were performed on a Perkin Elmer 2400 Series II CHNS/O Elemental Analyzer (Waltham, MA, USA), using the classical Pregl–Dumas method, and considering an accepted deviation of elemental analysis results from the calculated one of 0.4%.

The FTIR spectra were recorded with a Bruker Vertex-70 spectrophotometer (Bruker Corporation, Billerica, MA, USA) with an optical system with diamond. The IR bands were given as w (weak), m (medium), s (intense), and vs (very intense) and were obtained using attenuated total reflection Fourier-transformed infrared (ATR-FTIR) spectra at room temperature.

NMR spectra were recorded on a Varian Unity Inova 400 instrument operating at 400 MHz for ^1^H, 100 MHz for ^13^C, and 376.3 MHz for ^19^F-NMR (Varian Medical Systems, Palo Alto, CA, USA). As solvent, we used perdeuterated dimethyl sulfoxide with a min. 98% isotopic purity. The spectra were recorded at a temperature of 20 ± 1 °C. In order to perform the right attribution of spectral signals, we used different techniques to simplify the spectra, like the homonuclear decoupling and the specific deuteration. 

The 2-(4-ethyl-phenoxymethyl)benzoic acid (**2**) and 2-(4-ethyl-phenoxymethyl)benzoyl chloride (**3**) derivatives were obtained with good yields, following the method described in [55].

#### 4.1.1. General Synthesis Procedure of the New Benzoylthiourea Derivatives

To a solution of ammonium thiocyanate (0.01 mol) in acetone (5 mL) we added a solution of 2-(4-ethylphenoxymethyl)benzoyl chloride (**3**) (0.01 mol) in acetone (15 mL) to obtain in situ arylisothiocyanate (**4**). The reaction mixture was refluxed for 1 h, and then cooled at room temperature. A primary amine solution (0.01 mol) in acetone (2 mL) was added to the mixture and refluxed for 1 h. After cooling, the compound was precipitated and poured into 500 mL cold water, the solid was filtered, dried, and then recrystallized from isopropanol with active carbon.

##### *2-((4-Ethylphenoxy)methyl)-N-(3-fluorophenylcarbamothioyl)benzamide* (**5a**), 3.18 g white crystals (yield 78%), mp 109–110 °C.

^1^H-NMR (DMSO-d6): 12.52 (br s, 1H, NH), 11.92 (br s, 1H, NH), 7.70 (dt, ^4^*J*_(H18–H20,22)_ = 2.2 Hz, ^3^*J*_(F–H18)_ = 10.9 Hz, 1H, H-18), 7.62 (dd, *J* = 1.2 Hz, *J* = 7.6 Hz, 1H, H-7), 7.60 (m, 1H, H-4), 7.58 (td, *J* = 1.4 Hz, *J* = 7.4 Hz, 1H, H-5), 7.47 (td, *J* = 1.4 Hz, *J* = 7.5 Hz, 1H, H-6), 7.43 (td, ^4^*J*_(H21–F)_ = 6.6 Hz, ^3^*J*_(H21–H20,22)_ = 8.2 Hz, 1H, H-21), 7.34 (ddd, ^4^*J*_(H22–H20)_ = 1.0 Hz, ^4^*J*_(H18–H22)_ = 2.2 Hz, ^3^*J*_(H22–H21)_ = 8.2 Hz, 1H, H-22), 7.09 (tdd, ^4^*J*_(H22–H20)_ = 1.0 Hz, ^4^*J*_(H18–H20)_ = 2.2 Hz, ^3^*J*_(H20–H21)_ = 8.2 Hz, ^3^*J*_(H20–F)_ = 8.2 Hz, 1H, H-20), 7.08 (d, *J* = 8.6 Hz, 2H, H-11, H-13), 6.89 (d, *J* = 8.6 Hz, 2H, H-10, H-14), 5.28 (s, 2H, H-8), 2.50 (q, *J* = 7.5 Hz, 2H, H-15), 1.12 (t, *J* = 7.5 Hz, 3H, H-15’) (Appendix A).

^13^C-NMR (DMSO-d6): 179.06 (C-16), 170.13 (C-1), 161.62 (d, *J*_(F-C19)_ = 241.3 Hz, C-19), 156.28 (C-9), 139.47 (d, ^3^*J*_(F-C17)_ = 10.6 Hz, C-17), 136.18 (Cq), 135.86 (Cq), 133.21 (Cq), 131.01 (C-5), 130.16 (d, *J*_(F-C21)_ = 9.6 Hz, C-21), 128.56 (C-11, C-13), 128.45 (C-4), 128.33 (C-7), 127.69 (C-6), 120.11 (d, ^4^*J*_(F-C22)_ = 2.9 Hz, C-22), 114.53 (C-10, C-14), 112.84 (d, *J*_(F-C20)_ = 20.9 Hz, C-20), 111.03 (d, *J*_(F-C18)_ = 25.6 Hz, C-18), 67.49 (C-8), 27.23 (C-15), 15.70 (C-15’) (Appendix A).

^19^F-NMR (DMSO-d6): −112.59 (dd, F, ^3^*J*_(H20–F)_ = 8.2 Hz, ^3^*J*_(F–H18)_ = 10.9 Hz).

FTIR (solid in ATR, ν cm^−1^): 3163s, 3028m, 2966m, 2934m, 2881m, 1688m, 1602m, 1514vs, 1452s, 1378m, 1300m, 1273m, 1245s, 1172s, 1139s, 1076m, 1013m, 982w, 897w, 860w, 827m, 777w, 723m, 643w, 608w.

Anal. Calcd for C_23_H_21_FN_2_O_2_S (408.49): C, 67.63; H, 5.18; N, 7.83; S, 7.85%. Found: C, 67.31; H, 4.97; N, 7.74; S 8.01%.

##### *2-((4-Ethylphenoxy)methyl)-N-(2,3,4-trifluorophenylcarbamothioyl)benzamide* (**5b**) 3.37 g white crystals (yield 76%), mp 107–108 °C.

^1^H-NMR (DMSO-d6): 12.14 (br s, 1H, NH), 12.11 (br s, 1H, NH), 7.63 (dd, *J* = 1.2 Hz, *J* = 7.61 Hz, 1H, H-7), 7.61- 7.51 (m, 3H, H-4, H-5, H-22), 7.47 (td, *J* = 1.4 Hz, *J* = 7.5 Hz, 1H, H-6), 7.35 (td, ^4^*J*_(H21–F19)_ = 2.2 Hz, ^3^*J*_(H21–F20)_= 9.4 Hz, ^3^*J*_(H21–H22)_ = 9.4 Hz, 1H, H-21), 7.08 (d, *J* = 8.6 Hz, 2H, H-11, H-13), 6.89 (d, *J* = 8.6 HZ, 2H, H-10, H-14), 5.27 (s, 2H, H-8), 2.50 (q, *J* = 7.5 Hz, 2H, H-15), 1.13 (t, *J* = 7.5 Hz, 3H, H-15’) (Appendix A).

^13^C-NMR (DMSO-d6): 181.10 (C-16), 170.13 (C-1), 156.22 (C-9), 148.72 (ddd, ^3^*J*_(C18–F20)_ = 2.6 Hz, ^2^*J*_(C18–F19)_ = 9.9 Hz, *J*_(C–F)_ = 245.7 Hz, C-18), 145.47 (ddd, ^3^*J*_(C20–F18)_ = 3.6 Hz, ^2^*J*_(C20–F19)_ = 9.9 Hz, *J*_(C20–F20)_ = 249.7 Hz, C-20), 139.23 (dt, ^2^*J*_(C19–F18,20)_ = 9.9 Hz, *J*_(C19–F19)_ = 246.8 Hz, C-19), 136.23 (Cq), 135.87 (Cq), 133.08 (Cq), 131.07 (C-5), 128.56 (C-11, C-13), 128.42 (C-4, C-7), 127.72 (C-6), 124.27 (dd, ^3^*J*_(C17–F19)_ = 3.7 Hz, ^2^*J*_(C17–F18)_ = 9.1 Hz, C-17), 122.71 (dd, ^3^*J*_(C22–F18)_ = 3.7 Hz, ^3^*J*_(C22–F20)_ = 8.1 Hz, C-22), 114.59 (C-10, C-14), 111.62 (dd, ^3^*J*(C^21^-F^19^) = 3.7 Hz, ^2^*J*(C^21^-F ^20^) = 17.6 Hz, C-21), 67.51 (C-8), 27.25 (C-15), 15.77 (C-15’) (Appendix A).

^19^F-NMR (DMSO-d6): −161.03 (dq, ^4^*J*_(F20–F18)_ = 6.0 Hz, ^4^*J*_(F20–H22)_ = 6.0 Hz, ^3^*J*_(F20–F19)_ = 21.8 Hz, F-20), −140.50 (dt, ^3^*J*_(F18–F20)_ = 6.0 Hz, ^3^*J*_(F18–H22)_ = 6.0 Hz, ^3^*J*_(F18–F19)_ = 21.8 Hz, F-18), −136.78 (td, ^4^*J*_(H21–F19)_ = 9.4 Hz, ^3^*J*_(F19–F18,20)_ = 21.8 Hz, F-19).

FTIR (solid in ATR, ν cm^−1^): 3155m, 3064m, 2970m, 2873w, 1691s, 1616w, 1506vs, 1381w, 1315m, 1256m, 1215m, 1174s, 1070s, 1045m, 994m, 907w, 875w, 826w, 725m, 692m, 658w, 600w.

Anal. Calcd for C_23_H_19_F_3_N_2_O_2_S (444.47): C, 62.15; H, 4.31; N, 6.30; S, 7.21%; Found: C, 62.29; H, 4.28; N, 6.29; S 7.30%.

##### *2-((4-Ethylphenoxy)methyl)-N-(2,4,5-trifluorophenylcarbamothioyl)benzamide* (**5c**) 3.24 g white crystals (yield 73%), mp 114–115 °C.

^1^H-NMR (DMSO-d6): 12.30 (br s, 1H, NH), 12.12 (br s, 1H, NH), 8.07 (ddd, ^3^*J*_(H22–F18)_ = 7.2 Hz, ^3^*J*_(H22–F20)_ = 8.6 Hz, ^3^*J*_(H22–F21)_ = 11.7 Hz, 1H, H-22), 7.70 (td, ^3^*J*_(H19–F21)_ = 7.4 Hz, ^3^*J*_(H19–F20, 21)_ = 10.7 Hz, 1H, H-19), 7.62 (dd, *J* = 1.2 Hz, *J* = 7.6 Hz, 1H, H-7), 7.60 (m, 1H, H-4), 7.57 (td, *J* = 1.2 Hz, *J* = 7.6 Hz, 1H, H-5), 7.47 (td, *J* = 1.4 Hz, *J* = 7.5 Hz, 1H, H-6), 7.08 (d, *J* = 8.6 Hz, 2H, H-11, H-13), 6.89 (d, *J* = 8.6 Hz, 2H, H-10, H-14), 5.26 (s, 2H, H-8), 2.50 (q, *J* = 7.5 Hz, 2H, H-15), 1.12 (t, *J* = 7.5 Hz, 3H, H-15’) (Appendix A).

^13^C-NMR (DMSO-d6): 181.36 (C-16), 170.27 (C-1), 156.17 (C-9), 151.17 (ddd, ^4^*J*_(C18–F)_ = 2.6 Hz, ^3^*J*_(C18–F)_ = 9.9 Hz, *J*_(C–F)_ = 246.4 Hz, C-18), 147.17 (ddd, *J*_(C20–F)_ = 12.2 Hz, *J*_(C20–F)_ = 14.4 Hz, *J*_(C20–F)_ = 248.0 Hz, C-20), 144.89 (ddd, ^4^*J*_(C21–F18)_ = 3.7 Hz,^2^*J*_(C21–F20)_ = 13.3 Hz, *J*_(C21–F)_ = 242.0 Hz, C-21), 136.17 (Cq), 135.82 (Cq), 133.04 (Cq), 131.06 (C-5), 128.50 (C-11, C-13), 128.49 (C-4), 128.40 (C-7), 127.71 (C-6), 122.87 (ddd, ^4^*J*_(C17–F20)_ = 4.0 Hz, ^3^*J*_(C17–F21)_ = 9.1 Hz, ^2^*J*_(C17–F18)_ = 13.2 Hz, C-17), 115.29 (dd, ^3^*J*_(C22–F20)_ = 2.8 Hz, ^2^*J*_(C22–F21)_ = 23.5 Hz, C-22), 114.53 (C-10, C-14), 105.93 (dd, ^2^*J*_(C20–F)_ = 22.1 Hz, ^2^*J*_(C20–F)_ = 26.4 Hz,C-19), 67.48 (C-8), 27.18 (C-15), 15.68 (C-15’) (Appendix A).

^19^F-NMR (DMSO-d6): −118.49 (m, F-18), −131.70 (m, F-21), −137.65 (m, F-20).

FTIR (solid in ATR, ν cm^−1^): 3401w, 3269w, 3069w, 2964w, 1676m, 1602w, 1567s, 1508vs, 1439s, 1422m, 1381w, 1317s, 1239s, 1216s, 1170ms, 1151s, 1054w, 1036w, 882w, 869w, 834w, 752w, 737w, 680w, 642w, 613w.

Anal. Calcd for C_23_H_19_F_3_N_2_O_2_S (444.47): C, 62.15; H, 4.31; N, 6.30; S, 7.21%; Found: C, 61.89; H, 4.34; N, 6.27; S 7.26%.

##### *2-((4-Ethylphenoxy)methyl)-N-(2,4,6-trifluorophenylcarbamothioyl)benzamide* (**5d**) 3.20 g white crystals (yield 72%), mp 122–123 °C.

^1^H-NMR (DMSO-d6): 12.14 (br s, 1H, NH), 11.56 (br s, 1H, NH), 7.64 (dd, *J* = 1.2 Hz, *J* = 7.6 Hz, 1H, H-7), 7.60 (m, 1H, H-4), 7.57 (td, *J* = 1.2 Hz, *J* = 7.6 Hz, 1H, H-5), 7.47 (td, *J* = 1.42 Hz, *J* = 7.5 Hz, 1H, H-6), 7.30 (m, 2H, H-19, H-20), 7.08 (d, *J* = 8.6 Hz, 2H, H-11, H-13), 6.89 (d, *J* = 8.6 Hz, 2H, H-10, H-14), 5.27 (s, 2H, H-8), 2.51 (q, *J* = 7.5 Hz, 2H, H-15), 1.14 (t, *J* = 7.5 Hz, 3H, H-15’) (Appendix A).

^13^C-NMR (DMSO-d6): 182.27 (C-16), 169.75 (C-1), 160.82 (dt, ^3^*J*_(C20–F18,22)_ = 30.2 Hz *J*_(C20–F20)_ = 247.1 Hz, C-20), 158.15 (ddd, *J*_(C–F)_ = 7.3 Hz, *J*_(C–F)_ = 16.1 Hz, *J*_(C–F)_ = 250.0 Hz, C-18, C-22), 156.12 (C-9), 136.19 (Cq), 135.86 (Cq), 132.97 (Cq), 131.04 (C-5), 128.61 (C-4), 128.51 (C-11, C-13), 128.46 (C-7), 127.68 (C-6), 114.68 C-10, C-14), 113.35 (td, *J*_(C17–F20)_ = 5.1 Hz, *J*_(C17–F18, 22)_ = 16.5 Hz, C-17), 100.76 (td, *J*_(C19, 21–F20)_ = 3.6 Hz, *J*_(C19, 21–F18, 22)_ = 27.1 Hz, C-19, C-21), 67.53 (C-8), 27.21 (C-15), 15.74 (C-15’) (Appendix A).

^19^F-NMR (DMSO-d6): −108.85 (m, F-20), −114.85 (m, F-18, F-22).

FTIR (solid in ATR, ν cm^−1^): 3160m, 3066m, 3009m, 2965m, 2933m, 2872m, 1683s, 1644w, 1599w, 1508vs, 1449vs, 1384m, 1313w, 1246s, 1222m, 1173vs, 1126s, 1068w, 1041s, 1019m, 999m, 949w, 882w, 838m, 809w, 763m, 748m, 730m, 660m, 611w, 552w.

Anal. Calcd for C_23_H_19_F_3_N_2_O_2_S (444.47): C, 62.15; H, 4.31; N, 6.30; S, 7.21%; Found: C, 62.32; H, 4.38; N, 6.37; S 7.21%.

##### *2-((4-Ethylphenoxy)methyl)-N-(3,4,5-trifluorophenylcarbamothioyl)benzamide* (**5e**) 3.33 g light yellow crystals (yield 75%), mp 133–134 °C.

^1^H-NMR (DMSO-d6): 12.39 (br s, 1H, NH), 12.00 (br s, 1H, NH), 7.62- 7.58 (m, 1H, H-4), 7.64 (dd,

*J* = 1.2 Hz, *J* = 7.6 Hz, 1H, H-7), 7.57 (td, *J* = 1.2 Hz, *J* = 7.6 Hz, 1H, H-5), 7.47 (td, *J* = 1.4 Hz, *J* = 7.5 Hz, 1H, H-6), 7.09 (d, *J* = 8.6 Hz, 2H, H-11, H-13), 6.89 (d, *J* = 8.6 Hz, 2H, H-10, H-14), 5.26 (s, 2H, H-8), 2.51 (q, *J* = 7.5 Hz,2H, H-15), 1.12(t, *J* = 7.5 Hz, 3H, H-15’) (Appendix A).

^13^C-NMR (DMSO-d6): 179.60 (C-16), 169.99 (C-1), 156.24 (C-9), 146.69 (ddd, ^3^*J*_(C19,21–F19,21)_ = 5.4 Hz, ^2^*J*_(C19,21–F20)_ = 10.1 Hz, *J*_(C19,21–F19,21)_ = 245.0 Hz, C-19, C-21), 136.73 (dt, *J*_(C20–F19,21)_ = 15.7 Hz, *J*_(C20–F20)_ = 248.1 Hz, C-20), 136.17 (Cq), 135.87 (Cq), 134.06 (td, *J*_(C17–F20)_ = 4.0 Hz, *J*_(C17–F19,21)_ = 11.5 Hz, C-17), 133.11(Cq), 131.07 (C-5), 128.54 (C-11, C-13), 128.41 (C-4), 128.35 (C-7), 127.71 (C-6), 114.49 (C-10, C-14), 109.60 (m, ^3^*J*_(C18,22–F20)_ = 6.5 Hz, ^2^*J*_(C18,22–F19,21)_ = 23.4 Hz, C-18, C-22), 67.42 (C-8), 27.19 (C-15), 15.66 (C-15’) (Appendix A).

^19^F-NMR (DMSO-d6): −130.84 (dd, ^2^*J*_(H18,22–F19,21)_ = 9.8 Hz, ^3^*J*_(F20–F19,21)_ = 22.4 Hz, F-19, F-21), −159.53 (tt, ^4^*J*_(F20–H18,22)_ = 6.3 Hz, ^3^*J*_(F19,21–F20)_ = 22.4 Hz, F-20).

FTIR (solid in ATR, ν cm^−1^): 3174 m, 3038 m, 2965 m, 2879m, 2843w, 1689m, 1626w, 1609m, 1582w, 1554m, 1510vs, 1441s, 1368m, 1305m, 1252m, 1237s, 1211s, 1178vs, 1142s, 1079w, 1047s, 1015m, 864w, 850w, 832m, 820w, 753m, 715m, 654w, 591w.

Anal. Calcd for C_23_H_19_F_3_N_2_O_2_S (444.47): C, 62.15; H, 4.31; N, 6.30; S, 7.21%; Found: C, 62.34; H, 4.37; N, 6.21; S 7.15%.

##### *2-((4-Ethylphenoxy)methyl)-N-(2-trifluoromethylphenylcarbamothioyl)benzamide* (**5f**) 3.39 g white crystals (yield 74%); mp 123–124 °C.

^1^H-NMR (DMSO-d6): 12.44 (br s, 1H, NH), 12.14 (br s, 1H, NH), 7.79 (dq, ^4^*J*_(3F–H19)_ = 0.9 Hz, *J*_(H19–H20)_ = 8.4 Hz, 1H, H-19), 7.76-7.70 (m, 2H, H-21, H-22), 7.63 (dd, *J* = 1.2 Hz, *J* = 7.6 Hz, 1H, H-7), 7.61 (m, 1H, H-4), 7.57 (td, *J* = 1.2 Hz, *J* = 7.6 Hz, 1H, H-5), 7.53 (m, 1H, H-20), 7.47 (td, *J* = 1.4 Hz, *J* = 7.5 Hz, 1H, H-6), 7.09 (d, *J* = 8.6 Hz, 2H, H-11, H-13), 6.89 (d, *J* = 8.6 Hz, 2H, H-10, H-14), 5.27 (s, 2H, H-8), 2.52 (q, *J* = 7.5 Hz, 2H, H-15), 1.13 (t, *J* = 7.5 Hz, 3H, H-15’) (Appendix A).

^13^C-NMR (DMSO-d6): 181.39 (C-16), 170.61 (C-1), 156.16 (C-9), 136.17 (Cq), 135.77 (Cq), 135.69 (q, ^3^*J*_(3F-C17)_ = 2.2 Hz, C-17), 133.08 (Cq), 132.62 (C-22), 131.08 (C-5), 130.56 (C-20), 128.5 3(C-11, C-13), 128.49 (C-4), 127.76 (C-7), 127.62 (C-6), 126.09 (q, ^3^*J*_(3F-C20)_ = 5.2 Hz, C-19), 124.41(q, *J*_(3F-C18)_ = 29.3 Hz, C-18), 123.30 (q, *J*_(3F-C)_ = 271.7 Hz, CF_3_), 114.51 (C-10, C-14), 67.43 (C-8), 27.22 (C-15), 15.73 (C-15’) (Appendix A).

^19^F-NMR (DMSO-d6): −55.64 (3F).

FTIR (solid in ATR, ν cm^−1^):3176s, 3011m, 2972m, 2912m, 2870w, 1695m, 1608m, 1586m, 1508vs, 1471m, 1455m, 1374w, 1318s, 1284m, 1250s, 1230s, 1156vs, 1135s, 1107s, 1076m, 1059m, 1025s, 890w, 871w, 824m, 801w, 777m, 756s, 735m, 716w, 688w, 664w, 650w, 638w, 610w.

Anal. Calcd for C_24_H_21_F_3_N_2_O_2_S (458.49): C, 62.87; H, 4.62; N, 6.11; S, 6.99%; Found: C, 62.98; H, 4.58; N, 6.23; S 7.04%.

##### *2-((4-Ethylphenoxy)methyl)-N-(4-trifluoromethylphenylcarbamothioyl)benzamide* (**5g**) 3.25 g light yellow crystals (yield 71%), mp 138-139 °C.

^1^H-NMR (DMSO-d6): 12.59 (br s, 1H, NH), 11.96 (br s, 1H, NH), 7.88 (d, *J* = 8.4 Hz, 2H, H-19, H-21), 7.76 (d, *J* = 8.4 Hz, 2H, H-18, H-22), 7.63 (dd, *J* = 1.4 Hz, *J* = 7.2 Hz, 1H, H-7), 7.60 (dd, *J* = 1.7 Hz, *J* = 7.8 Hz, 1H, H-4), 7.57 (td, *J* = 1.4 Hz, *J* = 7.62 Hz, 1H, H-5), 7.47 (td, *J* = 1.7 Hz, *J* = 7.5 Hz, 1H, H-6), 7.07 (d, *J* = 8.6 Hz, 2H, H-11, H-13), 6.89 (d, *J* = 8.6 Hz, 2H, H-10, H-14), 5.28 (s, 2H, H-8), 2.50 (q, *J* = 7.5 Hz, 2H, H-15), 1.10(t, *J* = 7.5 Hz, 3H, H-15’) (Appendix A).

^13^C-NMR (DMSO-d6): 178.22 (C-16), 170.07 (C-1), 156.23 (C-9), 141.48 (Cq), 131.02 (C-5), 128.54 (C-11, C-13), 128.44 (C-7), 128.30 (C-4), 127.67 (C-6), 126.10 (q, *J*_(3F-C20)_ = 32.0 Hz, C-20), 125.65 (q, ^3^*J*_(3F-C19,21)_ = 3.7 Hz, C-19, C-21), 124.02 (q, *J*_(3F-C)_ = 271.0 Hz, CF_3_), 124.41 (C-18, C-22), 114.51 (C-10, C-14), 67.45 (C-8), 27.18 (C-15), 15.67 (C-15’) (Appendix A).

^19^F-NMR (DMSO-d6): −55.53 (3F).

FTIR (solid in ATR, ν cm^−1^): 3185m, 3028m, 2959m, 2930m, 2872w, 1691s, 1601m, 1553s, 1525vs, 1510vs, 1554m, 1414w, 1392w, 1319vs, 1239s, 1159vs, 1120vs, 1106vs, 1078w, 1066vs, 1032m, 1014m, 874w, 826m, 739m, 713m, 678w, 649w, 615w.

Anal. Calcd for C_24_H_21_F_3_N_2_O_2_S (458.49): C, 62.87; H, 4.62; N, 6.11; S, 6.99%; Found: C, 62.68; H, 4.68; N, 6.24; S 7.08%.

#### 4.1.2. Single Crystal X-ray Diffraction, Crystal Data Collection and Refinement

A colorless block crystal of C_23_H_19_F_3_N_2_O_2_S with approximate size of 0.400 × 0.350 × 0.200 mm was measured with a Rigaku R-AXIS RAPID II diffractometer using graphite monochromated Mo-K*_α_* radiation. The data were collected at a temperature of 20 ± 1 °C to a maximum 2θ value of 55.0°. Of the 15,071 reflections that were collected, 4927 were unique (R_int_ = 0.0251). The linear absorption coefficient, µ, for Mo-K*_α_* radiation is 1.983 cm^−1^. The data were corrected for Lorentz and polarization effects. The structure was solved by direct methods [54] and expanded using Fourier techniques. Refinement of F^2^ was done against all reflections. The weighted R factor, wR, and goodness of fit are based on F^2^. The non-hydrogen atoms were refined anisotropically. Hydrogen atoms were refined using the riding model. All calculations were performed using the CrystalStructure [56] crystallographic software package except for refinement, which was performed using SHELXL97 [57]. Crystallographic data were submitted to the Cambridge Structural Data Base with the deposition number 1965014.

### 4.2. Computational Analyses

The ligands were prepared using SPARTAN’14 software package [Spartan’14 Wavefunction, Inc. Irvine, CA]. The DFT/B3LYP/6-31 G^*^ level of basis set was used for the computation of molecular structure, vibrational frequencies, and energies of optimized structures. Molecular docking approach using CLC Drug Discovery Workbench Software was conducted.

All investigated compounds were virtually docked against *E. coli* DNA gyrase B, which was downloaded from the Protein Data Bank (PDB ID: 4DUH) [58].

The steps to go through to explore protein–ligand interaction are: Set up the binding site in a molecule project, dock the ligands imported to a molecule table, and inspect the docking results.

### 4.3. In Vitro Assessment of the Antimicrobial Activity of the Newly Synthesized F-Benzoylthiourea Derivatives on Planktonic Microbes (Grown in Suspension)

The inhibitory activity upon microbial growth was tested on Gram-positive (*S. aureus* ATCC 25923, *E. faecalis* ATCC 29212, *B. subtilis* ATCC 6633) and Gram-negative (*E. coli* ATCC 25922, *P. aeruginosa* ATCC 27853) bacteria and one fungal (*C. albicans* ATCC 10231) strain. The compounds were solubilized in DMSO to a final concentration of 10 mg/mL and further sterilized by filtration (using 0.22 µm filter membranes). The quantitative assay of the minimal inhibitory concentration (MIC, μg/mL) was performed by the microdilution method in 96 multi-well plates [59]. In this purpose, serial binary dilutions (ranging between 1024 and 8 μg/mL) were achieved in nutrient broth/YPG (yeast peptone glucose) and each well was seeded with microbial inoculum of 0.5 McFarland density. The plates were incubated for 24 hours at 37 °C in the case of bacterial strains, and 48 hours at 28 °C in the case of the fungal strain. The respective MIC values were reported as being the lowest concentration of the tested compound which inhibited the visible microbial growth.

### 4.4. Antimicrobial Activity of Newly Synthesized F-Benzoylthiourea Derivatives on Microbial Adherence and Biofilms’ Development on the Inert Substratum

The antibiofilm activity was assessed by the microtiter method. For this purpose, after reading the MIC, the microplates were emptied and the microbial biofilms formed on the plastic walls were fixed with cold methanol, stained with 1% violet crystal solution, and resuspended in 30% acetic acid. The absorbance of the colored suspension was measured at 490 nm with an ELISA reader (Apollo LB 911). The inhibition of the microbial biofilm development was interpreted as the decrease of the absorbance below the value registered for the positive control [60,61,62,63,64].

## 5. Conclusions

We report here the synthesis, characterization, and bioevaluation of new fluorinated compounds as potential antimicrobials. The antibacterial and antifungal activity was correlated with the position of fluorine or trifluoromethyl substituents on the phenyl ring. The molecular docking simulation was performed to predict the binding modes of the obtained benzoylthiourea derivatives to *E. coli* DNA gyrase B (chosen because *E. coli* strain was the most susceptible to the highest number of the tested compounds), the binding affinities, and the orientation. The docking score, predicting the antimicrobial activity of the studied compounds, was correlated with the experimental data obtained from the evaluation of the antimicrobial activity against the *E. coli* strain.

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
