# Peer review of "New Substituted Benzoylthiourea Derivatives: From Design to Antimicrobial Applications"

_molecules, 2020, doi:10.3390/molecules25071478_

Round 1

Reviewer 1 Report

The study of Limban et al. on substituted benzoylthiourea derivatives as antimicrobial agents is well conducted and increases the knowledges about design, SAR e modeling of the treated topic. In my opinion, there are no major observations to be made but the minor revisions reported in the file attached are requested. Some of the evidenced corrections are necessary in the Supplementary also.

Author Response

First of all, we would like to thank the reviewer for the very constructive observations that certainly have contributed to increasing the quality of our article.

First, we have decided to change the title, the present one being in our opinion more appropriate to the topics covered.

The nee title is: New substituted benzoylthiourea derivatives: from design to antimicrobial applications

We have made all the changes suggested by the reviewer. They are highlighted with Track changes.

Reviewer 2 Report

Manuscript number: molecules-723213

“New substituted benzamides - from design to antimicrobial applications”

The Article should be rejected because:

  1. The manuscript is not distinctly written. Specifically:
  • The title focuses on design and antimicrobial applications of novel benzamides, while in the text there are no considerations on the rational and the strategy of the design. Also, previous background on similar molecules is missing.
  • The Introduction is a mash up of different topics with no correlation one to the other. It starts with a few literatures of Organofluorine compounds as commercial drugs, with no references on research literature data, which could be like the object of the manuscript. The Introduction then soon moves to tell about thiourea derivatives, avoiding any connection with Organofluorine compounds mentioned before. The last, general and vague sentence of the Introduction would summarize the aim of the paper, without achieving its purpose because of its haziness.
  • The Chemistry is poor, intermediates 1,2 and 3 were already previously described and characterized. The novelty is limited to the seven final compounds (5a-5g), which are fully characterized in the manuscript, using arguable techniques. For example, the comparison of the IR spectra of the 7 derivatives is not so significant and the detailed analysis of IR is trivial and scholastic. NMR spectra are detailed, with an almost complete definition of each H and C (authors did not identify the two NH). Nevertheless, author mentioned they used GCOSY, GHMBS, GHSQC but none of their spectra was included in SI. Similarly, also the 19F of the seven compounds should be attached in the SI.
  • The Computational section is chaotic, which are the aims of this part? Tables 1, 2 and 3 could be shifted in SI, Figures 5 is also present in the SI with a different numeration. Which is the reason of this double ubiquity? The manuscript should include any Figure or Table that is essential for a clear and correct reading. I am not sure Figure 7 has a real significance; it could be translated into a list of numbers into the text. Moreover, Table 4 is confusing, if the authors want to highlight the features crucial for the validation of Lipinski rule, all the other data should be deleted.
  • Antimicrobial MICs are presented in a Figure and not in a table with defined values. It is completely disordered and reading the discussion is difficult without understanding the MIC values. Moreover, it seems the most active compounds have MIC of 100 μg/mL (more than 200 μM) which is a definitely high value for considering them active as antimicrobials.
  • In the Cytotoxicity assay authors used a concentration of 10 μg/mL which is really lower than the antimicrobial one. Why this choice? Usually the concentration used is equal or higher if compared to the active one. Furthermore, they did not quantify cell viability, neither include a microscopy images of untreated cells. How is the viability defined?

-The manuscript is full of grammatical errors and mistakes in English uses (RMN instead of NMR, all along pages 17-21, or Fluor instead of Fluorine in the Abstract at line24, as examples); the authors should entirely revise the manuscript, may aided by an English native speaker.

-All over the manuscript, Figures, Schemes and Tables are disorganized and not easily understandable. The author should not only cite them all along the text, but they should also conceive them in a better way, moving to the SI or including data in the text.

-The literature background of the manuscript is general and not focused on the topic.

-The results and the overall conclusions are not clearly.

Author Response

First of all, we would like to thank the reviewer for the very useful observations.

 We have hopefully addressed them all into account in the revised manuscript.

  • The title focuses on design and antimicrobial applications of novel benzamides, while in the text there are no considerations on the rational and the strategy of the design. Also, previous background on similar molecules is missing.

We have changed the title in order to better reflect the covered topics.

  • The Introduction is a mash up of different topics with no correlation one to the other. It starts with a few literatures of Organofluorine compounds as commercial drugs, with no references on research literature data, which could be like the object of the manuscript. The Introduction then soon moves to tell about thiourea derivatives, avoiding any connection with Organofluorine compounds mentioned before. The last, general and vague sentence of the Introduction would summarize the aim of the paper, without achieving its purpose because of its haziness.

 We reorganized the introduction, emphasizing the importance of pharmacophores present in various classes of medicines and reported by recent studies.

  • The Chemistry is poor, intermediates 1,2 and 3 were already previously described and characterized. The novelty is limited to the seven final compounds (5a-5g), which are fully characterized in the manuscript, using arguable techniques. For example, the comparison of the IR spectra of the 7 derivatives is not so significant and the detailed analysis of IR is trivial and scholastic. NMR spectra are detailed, with an almost complete definition of each H and C (authors did not identify the two NH). Nevertheless, author mentioned they used GCOSY, GHMBS, GHSQC but none of their spectra was included in SI. Similarly, also the 19F of the seven compounds should be attached in the SI.

We completed the Supplementary Material with the requested spectra.

  • The Computational section is chaotic, which are the aims of this part? Tables 1, 2 and 3 could be shifted in SI, Figures 5 is also present in the SI with a different numeration. Which is the reason of this double ubiquity? The manuscript should include any Figure or Table that is essential for a clear and correct reading. I am not sure Figure 7 has a real significance; it could be translated into a list of numbers into the text. Moreover, Table 4 is confusing, if the authors want to highlight the features crucial for the validation of Lipinski rule, all the other data should be deleted.

 We have reorganized the manuscriot and taking into account the reviewer' observations we have moved the tables 1, 2 and 3 to Supplementary Material

  • Antimicrobial MICs are presented in a Figure and not in a table with defined values. It is completely disordered and reading the discussion is difficult without understanding the MIC values.

Figure 9 was replaced by a Table presenting the MIC values

Moreover, it seems the most active compounds have MIC of 100 μg/mL (more than 200 μM) which is a definitely high value for considering them active as antimicrobials.

We agree that the MIC values might indicate a moderate antimicrobial activity, but however, taking into consideration their anti-biofilm activity, they still have a promising potential for the development of novel antimicrobial agents.

  • In the Cytotoxicity assay authors used a concentration of 10 μg/mL which is really lower than the antimicrobial one. Why this choice? Usually the concentration used is equal or higher if compared to the active one.

Because the stock solutions contained a high DMSO content, interfering with the cytotoxicity assay. Therefore, a further dilution of the stock solution had to be done.

  • Furthermore, they did not quantify cell viability, neither include a microscopy images of untreated cells. How is the viability defined?

A microscopy image of untreated cells was added for comparison.

The manuscript is full of grammatical errors and mistakes in English uses (RMN instead of NMR, all along pages 17-21, or Fluor instead of Fluorine in the Abstract at line24, as examples); the authors should entirely revise the manuscript, may aided by an English native speaker.

We thank for the consistent observations; we have revised the language of the manuscript and made the suggested corrections.

Reviewer 3 Report

The manuscript by Chifiriuc and coworkers reports the synthesis, characterization and bio-evaluation of seven  fluorinated  compounds as potential antimicrobials. Their synthetic steps are streigthforward and the characterization is great. The introduction is well referenced- The authors also reported their docking scores, cytotoxicity, etc. Although the compounds did't show excellent bioactivity they proved to be potential thiourea derivatives for further studies. The work is worth for publication after the following small corrections.

  1. Lines 45 -54 names several compounds. It will be more appealing if a scheme with their structure is presented together with their names, and bioactivity of selected compounds, (For an example, see Antibiotics 2019, 8, 178; doi:10.3390/antibiotics8040178)
  2. Scheme 1, add the reaction yields and reaction times to the scheme and re-draw the structures without writing the letters CH2, just draw bond line structures. Even just draw and(Et, instead of CH2CH3).
  3. Table 4. What Co-crystallized refers to?
  4. For docking studies and MIC studies, what or which was the control? it need to be included in figures 7 and 8.
  5. Figure 9, a microscopy image for untreated cells is missing. 
  6. For the characterization section, the color of each compound and the amount is missing, (critical to add that information)
  7. line 600. Don't need to add the all list of figures here.

Author Response

First of all, we would like to thank the reviewer for the consistent observations that helped us to improve our paper. 

  1. Lines 45 -54 names several compounds. It will be more appealing if a scheme with their structure is presented together with their names, and bioactivity of selected compounds, (For an example, see Antibiotics 2019, 8, 178; doi:10.3390/antibiotics8040178)

We introduced figure 1 according to the reviewer' suggestions.

  1. Scheme 1, add the reaction yields and reaction times to the scheme and re-draw the structures without writing the letters CH2, just draw bond line structures. Even just draw and(Et, instead of CH2CH3).

 We have made all the suggested changes.

 Table 4. What Co-crystallized refers to?

 After the changes made according to the suggestions of the other reviewers, table 4 became table 1 and we have added the name of co-crystallized (co-crystallyzed: 4-{[4'-methyl-2'-(propanoylamino)-4,5'-bi-1,3-thiazol-2-yl]amino}benzoic acid).

  1. For docking studies and MIC studies, what or which was the control? it need to be included in figures 7 and 8.

For the MIC studies the control was DMSO.

  1. Figure 9, a microscopy image for untreated cells is missing. 

The image for untreated cells was added.

  1. For the characterization section, the color of each compound and the amount is missing, (critical to add that information)

We added the color of each compound and the amount.

  1. line 600. Don't need to add the all list of figures here.

OK

Reviewer 4 Report

it seems to be a decent work combining synthesis and physical/calculation methods to asses the bioactivity of new compounds. I only suggest to improve the discussion about importance of fluorine-containing compounds in the pharmaceutical industry. The provided citations are very narrow and does not give the readers the idea of critical rile of fluorine in the design of bio-active compounds. There fore I suggest to cite the following articles and, as pointed out, increase the paragraph on the importance of fluorinated compounds. 

DOI:10.1002/chem.201901840;
DOI: 10.1021/acs.chemrev.5b00392. doi:10.1016/j.jfluchem.2014.06.026;

https://doi.org/10.3390/molecules25030482; https://doi.org/10.3390/molecules22060977 https://doi.org/10.3390/molecules22030483 https://doi.org/10.3390/molecules16086432 https://doi.org/10.3390/molecules25030745;  

Author Response

We thank the reviewer for the useful suggestions. We have studied the suggested articles and quoted them in the article. 

Round 2

Reviewer 2 Report

Manuscript number: molecules-723213

New substituted benzoylthiourea derivatives: from design to antimicrobial applications

The Article need a MAJOR REVISION because:

The manuscript was significantly improved, following previous suggestions and comments.

In particular, authors revised:

  • The title, that is now in line with the topic of the manuscript;
  • The introduction, which was better re-organized. Authors included an interesting and useful Figure; unfortunately, it is missing of important information, as the MICs on the reported strains. Authors should include them.
  • The Computational section, even if authors decided to keep Figure 8, which significance remains unknown for me. I suggest them just to mention the docking scores in the text.
  • Antimicrobial MICs, which were clearly listed in a Table. Unfortunately, these values are too high for being promising, and the comparison with DMSO is quite absurd. I suggest authors to delete DMSO column and to define the MIC of 5a-g as >128 or >256 (depending on DMSO cutoff; a higher value as no scientific meaning).
  • The language of the manuscript, which was satisfactory implemented.

The following sections require further implementations:

  • The Chemistry; The Scheme 1 should be re-written: the chemical structures are stretched, the atoms have different sizes, sometimes authors wrote CH2 and sometimes they used a line. Please equalize them, in order to let a better reading.
  • The Cytotoxicity assays. An analysis done like this is a serious nonsense and has no scientific meaning at all. If the MIC is generally in the range 128-256 μg/mL, you cannot use a concentration that is more than 10/20 times lower than the antimicrobial one and expect to have a cytotoxic effect!!! Authors must re-prepare the stock solutions, respecting the DMSO poisoning concentration, and perform the cytotoxicity assays following a rational protocol. Otherwise, they should completely delete this section.

Author Response

We are very grateful to the reviewer for the evaluation and comments; we have revised the manuscript accordingly and hope that the reviewer will consider that our article meets now the criteria required for publication.

  • The Scheme 1 should be re-written: the chemical structures are stretched, the atoms have different sizes, sometimes authors wrote CH2 and sometimes they used a line. Please equalize them, in order to let a better reading.
  • The Scheme 1 was re-written
  • The Computational section, even if authors decided to keep Figure 8, which significance remains unknown for me. I suggest them just to mention the docking scores in the text
  • In the Computational section, we decided to delete Figure 8 and to mention the docking scores in the text (line 301-307)
  • The authors included an interesting and useful Figure; unfortunately, it is missing of important information, as the MICs on the reported strains. Authors should include them.
  • Regarding Figure 1, our intention was to highlight the interest of other research groups in the thiourea structure and antimicrobial activities attributed to it. However, we cannot complete with MIC values because the respective researches were performed by using a qualitative, disk diffusion method.
  • Antimicrobial MICs, which were clearly listed in a Table. Unfortunately, these values are too high for being promising, and the comparison with DMSO is quite absurd. I suggest authors to delete DMSO column and to define the MIC of 5a-g as >128 or >256 (depending on DMSO cutoff; a higher value as no scientific meaning).
  • We have deleted the DMSO results and represented the MICs as >128 or >256, depending on DMSO cutoff; the results were re-discussed in accordance with the new data representation
  • The Cytotoxicity assays. 
  • We have chosen to delete this section, due to the impossibility to resume the experiments in the near future, due to the difficult COVID related emergency situation